# Prevalence and factors associated with depression, anxiety and stress symptoms among construction workers in Nepal

Bikram Adhikari[1]*, Lisasha Poudel[1], Niroj Bhandari[1], Nabin Adhikari[1], Bhawana Shrestha[1], Bikram Poudel[1], Anupama Bishwokarma[1], Bihari Saran Kuikel[2], Dinesh Timalsena[2], Bandana Paneru[2], Minani Gurung[3], Pramesh Koju[1], Rajendra Karkee[4], Anup Ghimire[4]

1 Research and Development Division, Dhulikhel Hospital, Kathmandu University Hospital, Dhulikhel, Nepal, 2 Department of Public Health and Community Programs, Kathmandu University School of Medical Sciences, Dhulikhel, Nepal, 3 One Health Research and Training Center, Kathmandu, Nepal, 4 School of Public Health and Community Medicine, B.P. Koirala Institute of Health Sciences, Dharan, Nepal

* bikram.adhikariadhitya@gmail.com

## Abstract

**Data Availability Statement:** All data files are publicly available in Open Science Framework database(URL: https://osf.io/xb8k5/).

## Introduction

The construction industry in Nepal, which employs a significant proportion of the population, ranks as one of the largest industries in the country. Construction work is physically demanding and can be risky due to the use of heavy machinery and the presence of intense physical labor. However, the physical and mental health of construction workers in Nepal is often neglected. This study aimed to assess psychological distress (depression, anxiety, and stress symptoms) and its association with socio-demographic, lifestyle, and occupational factors among construction workers in Kavre district, Nepal.

## Methods

We conducted a cross-sectional study from 1st October 2019 to 15th January 2020 among 402 construction workers in Banepa, and Panauti municipalities of Kavre district, Nepal. We collected data with face-to-face interviews using a structured questionnaire consisting of a) socio-demographic characteristics; b) lifestyle and occupational characteristics; and c) depression, anxiety and stress symptoms. We collected data using electronic forms in KoboToolbox and imported them into R version 3.6.2 for statistical analysis. We present parametric numerical variables as mean and standard deviation, and categorical variables as percentage and frequency. The confidence interval around proportion was estimated with the Clopper-Pearson method. We applied univariate and multivariable logistic regression to determine factors associated with depression symptoms, anxiety, and stress. The result of logistic regression was presented as crude odds ratio, adjusted odds ratio (AOR), and their 95% confidence interval (CI).

**Funding:** This study was not funded by any funding agencies.

**Competing interests:** The authors have declared that no competing interests exist.

## Results

The prevalence of depression, anxiety and stress symptoms were 17.1% (95%CI: 13.6–21.2), 19.2% (95%CI: 15.5–23.4) and 16.4% (95%CI: 12.9–20.4), respectively. In multivariable logistic regression analysis, depression symptom was positively associated with poor sleep quality (AOR = 3.51; 95%CI: 1.5–8.19; p-value: 0.004); stress symptom was positively associated with Brahmin ethnicity (AOR = 3.76; 95%CI:1.34–10.58; p-value: 0.012) and current smoking (AOR = 2.0; 95%CI: 1.11–3.82 p-value: 0.022). But anxiety symptoms were not associated with any of the variables.

## Conclusions

The prevalence of depression, anxiety, and stress symptoms were high among construction workers. Developing evidence-based and appropriate community-based mental health prevention programs among laborers and construction workers is recommended.

## Introduction

In Nepal, the construction industry is one of the biggest and most thriving industries where many Nepalese are employed [1]. In 2018, the sector employed 978,000 workers. Since then, the sector has been growing at an average rate of 9% each year. Therefore, in total, an estimated 1.2 million are employed in the construction industry [2].

Construction work is hard and often risky, as practiced in Nepal. It involves a high degree of mechanical skill and demanding physical labor [3]. Construction work includes members of specialist trades such as builders, electricians, carpenters, bricklayers, manual labor, armature fixing workers, internal finish workers and plumbers [1]. Construction workers are required to engage in highly repetitive and strenuous work with high job demand, long working hours, job insecurity and poor satisfaction. These factors have a significant impact on the physical and mental well-being of construction workers [4]. In addition, construction workers are frequently stressed over work-related accidents and injuries and often neglect to seek help, placing themselves at risk of further injuries and mental health problems, including depression symptoms, anxiety symptoms, and even suicide [5]. Mental health is a significant cause of suicide and disability globally, and substantial economic costs to the nation, and it has substantially affected the construction industry [4].

In Nepal, the causes for occupational hazards includes precarious working environment; crowded workplace; lack of supervision, monitoring, and training; close monitoring system negligence; workers' and employers' ignorance and recklessness; use of old or out-of-date machine or equipment; absence of adequate routine maintenance of tools, machine, and equipment; lack of standard safety equipment; breach of safety rules and unsuitable conditions [6]. Similarly, there is no incentive or disincentive to provide occupational health diagnosis and treatment facilities in the industries or to install safe and healthy equipment to replace worn-out and unsafe machinery, which causes stakeholders' reluctance [6, 7]. This causes distress for workers as they have to work in such conditions.

Due to the perceived inherent sources of hazards present in materials, processes, technologies, or products, construction sites are a perilous environment. It is essential to understand the physical health, mental health and well-being status of construction workers to reduce workplace injuries, prevent disabilities, and increase the nation's productivity [8, 9]. There are

limited studies to assess the mental health status of construction workers. In this context, we conducted a study to assess psychological distress (depression, anxiety, and stress symptoms), and their association with socio-demographic, lifestyle, and occupational factors among construction workers in Kavre district, Nepal.

## Methods

### Study design, population, and setting

We conducted a cross-sectional study from 1$^{st}$ October 2019 to 15$^{th}$ January 2020 among construction workers in Banepa and Panauti municipalities of Kavre district, Nepal. The current article is based on a study conducted among construction workers of Kavre district to assess their health status including low back pain, musculoskeletal pain, raised blood pressure and psychological distress (depression, anxiety and stress symptoms) [10].

### Sample size

We calculated sample size for this study by assuming a prevalence of depression symptoms of 21.3% [11], a 5% absolute error, a 95% confidence interval and a 10% non-response rate using the Cochrane formula. The calculated sample size was 284, but we ultimately recruited 402 construction workers using a convenience sampling method. A sample size of 402 was chosen to achieve the separate objective of our study to determine the prevalence of low back pain among construction workers [10].

### Sampling technique

We selected two municipalities (Panauti and Banepa) purposively for our study. We visited every ward in each municipality (12 wards from Panauti and 14 wards from Banepa) and located building construction sites with the assistance of local residents. We then approached the construction workers available at those sites and those who met the eligibility criteria were enrolled in the study. The inclusion criteria for the study were a) construction workers aged 18 years and above, and b) construction workers with work experience of one year or more. The exclusion criteria included: a) those construction workers who declined to give consent, and b) those with listening and speaking disabilities (due to problems in communication with sign languages).

### Ethical approval

We obtained ethical approval from the Institutional review committee of B.P. Koirala Institute of Health Sciences (Reference Number: 057/076/077-IRC, approval date: September 23, 2019) before conducting the study. We obtained oral informed consent from each participant before enrolling them in the study. We ensured that participation in the study was voluntary and maintained confidentiality and anonymity throughout the study.

### Data collection

The principal investigator collected data with face-to-face interviews using a structured questionnaire consisting of three sections: a) socio-demographic characteristics, b) lifestyle and occupational characteristics, and c) depression, anxiety, and stress symptoms. We collected data using electronic forms in KoboToolbox, a free and open-source online data entry tool developed by Harvard Humanitarian initiatives with support from various organizations like Brigham and Women's Hospital, United States Agency for International Development [12].

## Assessment of socio-demographic, lifestyle, and occupational factors

**Socio-demographic factors** included age (in years), gender (Male/Female), religion (Hindu/ Buddhism/ Others (Islam, Christian, Kirat)), ethnicity (Chhetri, Sanyasi/Brahmin/Janajati/ Other (Kami, Damai, Badi, Gaine, Madhesi Brahmin, Madhesi Janajati), marital status (married/ unmarried), and per capita income (dollar per family member per day).

**Lifestyle factors** included smoking status (current smoker/non-smoker), alcohol (current drinker/ non-drinker), sleep duration (in hours), and perceived sleep quality (good/intermediate/poor). Current smokers were defined as those who reported smoking any tobacco product within the last 30 days [13]. Current alcohol drinkers were those who consumed alcohol within the last 30 days [13]. Perceived sleep quality was assessed by asking participants to rate their sleep quality over the past year as "good", "intermediate", or "poor" [13].

**Occupational factors** included type of construction work (manual labor/bricklayer/internal finish worker/armature fixing/painter and electrician), working hours per week (in hours), job satisfaction (high/low), perceived job security (high/low), work-family life balance (high/low) and hostile work environment (yes/no). Job satisfaction was measured by asking participants if they "strongly agree", "agree", "disagree", or "strongly disagree" with the statement: "I am satisfied with my job". Responses, "strongly disagree" and "disagree", were defined as low job satisfaction [13]. The work-family imbalance was measured by asking if the participants "strongly agree", "agree", "neutral", "disagree", or "strongly disagree" with the statement "It is easy for me to combine work with family responsibilities." Responses, "strongly disagree" and "disagree", were defined as high work-family imbalance [13]. Exposure to the hostile work environment was measured by asking the participants if they were threatened, bullied, or harassed by anyone while they were on the job in the past 12 months. The response, "Yes", was defined as exposure to a hostile work environment [13]. Job insecurity was measured by asking participants if they "strongly agree", "agree", "disagree", or "strongly disagree" with the statement "I am worried about becoming unemployed." Responses, "strongly agree" and "agree", were defined as high job insecurity [13].

## Assessment of depression, anxiety, and stress symptoms

We used the Nepali version of the validated Depression Anxiety and Stress Scale-21 (DASS-21) [14] to assess depression, anxiety, and stress symptoms. The internal consistency of the DASS-21 Nepali version was 0.77 for DASS-depression; 0.80 for DASS-anxiety; and 0.82 for DASS-stress, which indicates Cronbach's alpha values. It has been used extensively in previous studies globally as well as in Nepal [15].

The DASS-21 scale contains 21 items (7 items each for depression, anxiety, and stress symptoms). Each participant is asked to score every item on a scale from 0 to 3, where 0 indicates "did not apply to me at all" and 3 indicates "apply to me at all". The total scores for depression, anxiety, and stress symptoms were calculated by summing the scores for each scale followed by multiplication with a factor of two [15]. We classified patients into mild, moderate, severe, and extremely severe conditions of depression, anxiety, and stress symptoms based on the total score within each subdomain [14]. Depression scores were classified as: normal (0–9), mild (10–13), moderate (14–20), severe (21–27), and extremely severe (27+). Anxiety score was classified as: normal (0–7), mild (8–9), moderate (10–14), severe (15–19), and extremely severe (20+). Stress score was classified as: normal (0–14), mild (15–18), moderate (19–25), severe (26–33), and extremely severe (34+). Depression, anxiety, and stress symptoms were further classified as binary (present/absent) if at least mild conditions were present.

## Statistical analysis

We imported data from the KoboToolbox platform [12] into R version 3.6.2 [16] for statistical analysis. Categorical variables were presented as percentage and frequency. We used the Clopper-Pearson method to calculate confidence intervals around the proportion. We present parametric numerical variables as mean and standard deviation. We applied univariate and multivariable logistic regression to determine factors associated with depression symptoms, anxiety, and stress symptoms. In our multivariable logistic regression analysis, we incorporated all variables from the univariate model. We presented univariate and multivariable logistic regression results using crude odds ratios (COR), adjusted odds ratios (AOR), their 95% confidence intervals and p-values. We conducted all tests at a 95% confidence level and considered p-values less than 0.05 to be statistically significant.

## Result

We assessed a total of 456 construction workers of which 415 were eligible to be included in the study. Thirteen construction workers declined to participate in the study and the response rate of the study was 96.9%. Of total 402 participants, 16.2% were females, and 83.8% were males. The age of the construction workers ranged from 18 to 64 years, with a mean age of 31.9±9.5 years. About three-fifths of the participants were Hindus (77.4%) followed by Buddhists (18.2%) and Kirat (3.0%). About 15.9% were literate and the remaining 84.0% had at least primary level education. About three-fifths (76.9%) of the participants were married. Of the total construction workers, 41.3% were manual labor, 25.9% were bricklayers, 14.7% were internal finish worker, 12.2% were armature fixing, 3.5% were painters and remaining 2.5% were electricians. The average working hours per week was 58.4±19.4. The majority of the construction workers had work experience of 2 to 10 years (61.2%) in the construction industry. The readers are referred elsewhere [10] for further details on socio-demographic characteristics, lifestyle characteristics and occupational characteristics of the construction workers,

### Prevalence of depression, anxiety, and stress

Table 1 presents the prevalence of depression, anxiety, and stress symptoms. The prevalence of depression, anxiety and stress symptoms were 17.1% (95% CI: 13.6–21.2), 19.2% (95% CI: 15.5–23.4) and 16.4% (95% CI: 12.9–20.4).

Table 2 presents univariate logistic regression results to determine factors associated with depression, anxiety, and stress symptoms. In univariate analysis, the odds of having depression symptoms were 2.37 (95% CI: 1.28–4.38; p-value: 0.006) times higher among females compared to males, 4.13 (95% CI: 1.99–8.56; p-value: <0.001) times higher among those who perceived poor sleep compared to those who perceived good sleep, and 1.99 (95% CI: 1.18–2.28; p-value: 0.010) times higher among those who perceived poor job security compared to those who perceived high job security. The odds of having anxiety symptoms were 1.99 (95% CI:

**Table 1.  Prevalence of depression anxiety and stress among construction workers in Kavre district, Nepal (N = 402).**

| Category | Depression symptoms % (95%CI) | Anxiety symptoms % (95%CI) | Stress symptoms % (95%CI) |
|---|---|---|---|
| **Mild** | 6.2 (3.4–11.0) | 4.0(1.9–8.2) | 11.2(7.5–16.3) |
| **Moderate** | 6.2(3.4–11.0) | 7.2(4.1–12.2) | 3.0(1.4–6.4) |
| **Severe** | 3.2(1.4–7.2) | 3.0(1.3–6.9) | 2.2(0.9–5.4) |
| **Extremely Severe** | 1.5(0.5–4.8) | 5.0(2.6–9.5) | - |

%: percent; CI: Confidence interval

**Table 2. Univariate logistic regression analysis to determine factors associated with depression, anxiety and stress symptoms among construction workers in Kavre district, Nepal.**

| Variables | Depression symptoms | | p value | Anxiety symptoms | | p value | Stress symptoms | | p value |
|---|---|---|---|---|---|---|---|---|---|
| | n (%) | COR (95%CI) | | n (%) | COR (95%CI) | | n (%) | COR (95%CI) | |
| **Age (in years),** mean±sd | 33.3±10.5 | 1.02(0.99–1.05) | 0.137 | 32.9±10.5 | 1.02(0.99–1.04) | 0.253 | 32.6±8.7 | 1.01(0.98–1.04) | 0.450 |
| **Gender** | | | | | | | | | |
| Male | 50 (14.8) | Ref | | 58(17.2) | Ref | | 51(15.1) | Ref | |
| Female | 19 (29.2) | **2.37(1.28–4.38)** | **0.006** | 19(29.2) | **1.99(1.09–3.64)** | **0.026** | 15(23.1) | 1.68(0.88–3.22) | 0.116 |
| **Religion** | | | | | | | | | |
| Hindu | 54(17.4) | Ref | | 62(19.9) | Ref | | 53(17.0) | Ref | |
| Non-Hindu | 15(16.5) | 0.94(0.50–1.76) | 0.185 | 15(16.5) | 0.79(0.43–1.47) | 0.462 | 13(14.3) | 0.81(0.42–1.57) | 0.533 |
| **Ethnicity** | | | | | | | | | |
| Chhetri/Sanyasi | 11(16.9) | Ref | | 13(20.0) | Ref | | 9(13.8) | Ref | |
| Brahmin | 13(23.6) | 1.52(0.62–3.73) | 0.361 | 16(29.1) | 1.64(0.71–3.81) | 0.249 | 16(29.1) | **2.55(1.02–6.36)** | **0.044** |
| Janajati | 41(16.1) | 0.94(0.45–1.95) | 0.869 | 43(16.9) | 0.81(0.41–1.62) | 0.553 | 36(14.1) | 1.02(0.47–2.25) | 0.955 |
| Other## | 4(14.8) | 0.85(0.25–2.96) | 0.803 | 5(18.5) | 0.91(0.29–2.86) | 0.870 | 5(18.5) | 1.41(0.44.69) | 0.571 |
| **Marital status** | | | | | | | | | |
| Married | 56(18.1) | Ref | | 58(18.8) | Ref | | 53(17.2) | Ref | |
| Unmarried | 13(14.0) | 0.73(0.38–1.41) | 0.354 | 19(20.4) | 1.11(0.62–1.98) | 0.721 | 13(14.0) | 0.79(0.41–1.51) | 0.470 |
| **Per capita income,** mean±sd | 2.7±1.0 | 0.918(0.74–1.15) | 0.451 | 2.8±1.24 | 0.97(0.84–1.13) | 0.724 | 2.9±1.6 | 0.99(0.87–1.13) | 0.910 |
| **Smoking** | | | | | | | | | |
| Non-smoker | 27(15.5) | Ref | | 44(19.3) | Ref | | 30(16.0) | Ref | |
| Current smoker | 42(18.4) | 0.81(0.48–1.38) | 0.445 | 33(19.0) | 0.98(0.59–1.62) | 0.933 | 34(16.8) | **1.72(1.01–2.93)** | **0.045** |
| **Alcohol intake** | | | | | | | | | |
| Non-drinker | 39(19.5) | Ref | | 40(20.0) | Ref | | 32(16.0) | Ref | |
| Current drinker | 30(14.9) | 0.72(0.42–1.21) | 0.218 | 37(18.3) | 0.90(0.55–1.48) | 0.897 | 34(16.8) | 1.06(0.63–1.80) | 0.822 |
| **Sleep duration** | 7.7±1.3 | 0.83(0.68–1.01) | 0.064 | 7.8±1.3 | 0.86(0.71–1.04) | 0.117 | 8.0±1.1 | 1.04(0.85–1.27) | 0.721 |
| **Perceived sleep quality** | | | | | | | | | |
| Good | 42(13.6) | Ref | | 50(16.2) | Ref | | 46(14.9) | Ref | |
| Intermediate | 12(21.4) | 1.73(0.84–3.53) | 0.135 | 14(25.0) | 1.72(0.87–3.38) | 0.116 | 11(19.6) | 1.39(0.67–2.89) | 0.374 |
| Poor | 15(39.5) | **4.13(1.99–8.56)** | **<0.001** | 13(34.2) | **2.68(1.28–5.60)** | **0.009** | 9(23.7) | 1.77(0.78–3.98) | 0.169 |
| **Type of construction work** | | | | | | | | | |
| Manual Labor | 35(21.1) | Ref | | 40(24.1) | Ref | | 30(18.1) | Ref | |
| Bricklayer | 15(14.4) | 0.63(0.33–1.22) | 0.173 | 18(17.3) | 0.66(0.36–1.23) | 0.188 | 15(14.4) | 0.76(0.39–1.50) | 0.434 |
| Other # | 19(14.4) | 0.63(0.34–1.16) | 0.138 | 19(14.4) | **0.53(0.29–0.97** | **0.039** | 21(15.9) | 0.86(0.47–1.58) | 0.623 |
| **Working hours per week,** mean±sd | 55.9±19.1 | 0.99(0.98–1.01) | 0.243 | 55.8±19.6 | 0.99(0.98–1.01) | 0.205 | 54.2±20.0 | 0.99(0.97–1.00) | 0.58 |
| **Job satisfaction** | | | | | | | | | |
| High | 48(15.6) | Ref | | 54(17.5) | Ref | | 44(14.3) | Ref | |
| Low | 21(22.3) | 1.56(0.88–2.77) | 0.130 | 23(24.5) | 1.52(0.88–2.65) | 0.137 | 22(23.4) | **1.83(1.03–3.26)** | **0.039** |
| **Perceived job security** | | | | | | | | | |
| Yes | 28(12.7) | Ref | | 32(14.5) | Ref | | 30(13.6) | Ref | |
| No | 41(22.5) | **1.99(1.18–2.28)** | **0.010** | 45(24.7) | **1.93(1.17–3.19)** | **0.011** | 36(19.8) | 1.56(0.92–2.65) | 0.100 |
| **Family work balance** | | | | | | | | | |
| Yes | 39(14.7) | Ref | | 48(18.0) | Ref | | 42(15.8) | Ref | |
| No | 30(22.1) | 1.65(0.97–2.80) | 0.064 | 29(21.3) | 1.23(0.73–2.06) | 0.430 | 24(17.6) | 1.14(0.66–1.98) | 0.634 |
| **Hostile work environment** | | | | | | | | | |
| No | 58(16.5) | Ref | | 67(19.1) | Ref | | 58(16.5) | Ref | |

(*Continued*)

**Table 2.** (Continued)

| Variables | Depression symptoms | | p value | Anxiety symptoms | | p value | Stress symptoms | | p value |
|---|---|---|---|---|---|---|---|---|---|
| | n (%) | COR (95%CI) | | n (%) | COR (95%CI) | | n (%) | COR (95%CI) | |
| Yes | 11(21.6) | 1.39(0.67–2.87) | 0.374 | 10(19.6) | 1.03(0.49–2.17) | 0.930 | 8(15.7) | 0.94(0.42–2.10) | 0.880 |

n: frequency; %: Percent; sd: standard deviation; CI: Confidence interval; COR: Crude Odds Ratio; AOR: Adjusted Odds Ratio; Ref: reference group.

\# Internal Finish Worker, Armature Fixing, Painter and Electrician

\#\# Kami, Damai, Badi, Gaine, Madhesi Brahmin, Madhesi Janajati

1.09–3.64; p-value: 0.026) times higher among females compared to males, 2.68 (95% CI: 1.28–5.60; p-value: 0.009) times higher among those who perceived poor sleep compared to those who perceived good sleep, and 1.93 (95% CI: 1.17–3.19; p-value: 0.011) times higher among those who perceived poor job security compared to those who perceived high job security. The odds of stress symptoms were 2.55 (95% CI: 1.02–6.36) times in Brahmin ethnic groups compared to Chhetri/Sanyasi ethnic groups, 1.72 (95% CI: 1.01–2.93; p-value: 0.045) times higher among current smokers compared to non-smokers, and 1.83 (95% CI: 1.03–3.26; p-value: 0.039) times higher among those with low job satisfaction compared to those with high job satisfaction.

In multivariable logistic regression analysis (Table 3), the odds of having depression symptoms were 3.51 times (95% CI: 1.5–8.19; p-value: 0.004) higher among those with perceived poor sleep quality compared to those who perceived good sleep quality after adjusting for age, gender, religion, ethnicity, marital status, per capita income, current alcohol consumption, current smoking, sleep duration, perceived sleep quality, type of construction workers, working hours per week, job satisfaction, job insecurity, family-work balance, and hostile work environment. The odds of having stress symptoms were 3.76 times (95% CI: 1.34–10.58; p-value: 0.012) higher among the Brahmin ethnic group compared to Chhetri/Sanyasi and 2.0 times (95% CI: 1.11–3.82; p-value: 0.022) higher among current smokers compared to non-smokers after adjusting for all sociodemographic, lifestyle, occupational factors. We did not find any association of anxiety symptoms with any of the variables.

## Discussion

The current study focused on the prevalence of depression, anxiety and stress symptoms and their association with socio-demographic, lifestyle, and occupational factors among construction workers in Kavre district, Nepal. The results in the present study indicate that the prevalence of depression symptoms, anxiety and stress were 17.1%, 19.2% and 16.4% respectively. Depression symptoms were positively associated with perceived poor sleep quality; stress symptoms were positively associated with Brahmin ethnicity and current smoking, but anxiety symptoms were not associated with selected variables.

The prevalence of depression and anxiety symptoms in our population was much higher than previously reported findings by WHO in the general population of Nepal with 3.2% depression and 3.6% anxiety [17]. Construction workers have a greater frequency of depression and anxiety symptoms compared to the general population, which might be linked to job-related stress, insecurity, dissatisfaction, financial strain, a hostile work environment and substances that are used as a coping mechanism for the hostile work environment, poor work environment, and stressful construction tasks [18]. Studies from the United Kingdom and Korea reported 2% to 11% prevalence of depression among construction workers, which was lower than this study [19, 20]. Soram Lim et al. reported 37.6% [21], and Lee M reported 25%

**Table 3. Multivariable logistic regression analysis to determine factors associated with depression symptoms, anxiety symptoms and stress among construction workers in Kavre district, Nepal.**

| Variables | Depression symptoms | | Anxiety symptoms | | Stress symptoms | |
|---|---|---|---|---|---|---|
| | AOR (95%CI) | p value | AOR (95%CI) | p value | AOR (95%CI) | p value |
| **Age (in years),** | 1.01(0.97–1.04) | 0.606 | 1.01(0.98–1.05) | 0.439 | 1(0.96–1.03) | 0.890 |
| **Gender** | | | | | | |
| Male | Ref | | Ref | | Ref | |
| Female | 1.33(0.52–3.38) | 0.554 | 1.6(0.64–3.95) | 0.313 | 1.65(0.61–4.49) | 0.325 |
| **Religion** | | | | | | |
| Hindu | Ref | | Ref | | Ref | |
| Non-Hindu | 1.32(0.6–2.94) | 0.492 | 1.01(0.47–2.18) | 0.982 | 0.83(0.37–1.87) | 0.655 |
| **Ethnicity** | | | | | | |
| Chhetri/Sanyasi | Ref | | Ref | | Ref | |
| Brahmin | 1.45(0.54–3.92) | 0.462 | 1.99(0.78–5.08) | 0.152 | **3.76(1.34–10.58)** | **0.012** |
| Janajati | 0.79(0.32–1.93) | 0.599 | 0.77(0.33–1.8) | 0.548 | 1.35(0.53–3.49) | 0.531 |
| Other## | 1.04(0.42–2.57) | 0.941 | 1.11(0.47–2.64) | 0.810 | 1.57(0.59–4.21) | 0.367 |
| **Marital status** | | | | | | |
| Married | Ref | | Ref | | Ref | |
| Unmarried | 0.99(0.45–2.18) | 0.973 | 1.81(0.88–3.74) | 0.109 | 1.08(0.49–2.39) | 0.852 |
| **Per capita Income** | 0.93(0.74–1.15) | 0.490 | 0.97(0.84–1.13) | 0.707 | 1.02(0.9–1.15) | 0.814 |
| **Smoking** | | | | | | |
| Non-smoker | Ref | | Ref | | Ref | |
| Current smoker | 0.86(0.47–1.59) | 0.632 | 0.98(0.54–1.76) | 0.938 | **2.06(1.11–3.82)** | **0.022** |
| **Alcohol intake** | | | | | | |
| Non-drinker | Ref | | Ref | | Ref | |
| Current drinker | 0.85(0.43–1.71) | 0.651 | 1.39(0.71–2.74) | 0.337 | 1.31(0.64–2.7) | 0.460 |
| **Sleep duration** | 0.9(0.73–1.11) | 0.342 | 0.89(0.73–1.09) | 0.263 | 1.09(0.88–1.36) | 0.430 |
| **Sleep quality** | | | | | | |
| Good | Ref | | Ref | | Ref | |
| Intermediate | 1.64(0.75–3.58) | 0.219 | 1.54(0.73–3.23) | 0.257 | 1.43(0.64–3.22) | 0.385 |
| Poor | **3.51(1.5–8.19)** | **0.004** | 1.95(0.83–4.56) | 0.125 | 1.44(0.56–3.69) | .444 |
| **Type of construction work** | | | | | | |
| Manual Labor | Ref | | Ref | | Ref | |
| Bricklayer | 1(0.46–2.16) | 0.998 | 1.6(0.77–3.36) | 0.210 | 0.86(0.39–1.88) | 0.704 |
| Other # | 0.66(0.3–1.49) | 0.322 | 0.96(0.45–2.09) | 0.927 | 0.77(0.35–1.71) | 0.526 |
| **Work hours per day** | 1(0.98–1.02) | 0.931 | 1(0.98–1.01) | 0.732 | 0.99(0.97–1) | 0.106 |
| **Job satisfaction** | | | | | | |
| High | Ref | | Ref | | Ref | |
| Low | 0.66(0.29–1.50) | 0.318 | 1.02(0.46–2.26) | 0.956 | 1.97(0.82–4.75) | 0.131 |
| **Perceived job security** | | | | | | |
| Yes | Ref | | Ref | | Ref | |
| No | 1.83(0.93–3.60) | 0.080 | 1.85(0.98–3.52) | 0.059 | 1.25(0.63–2.49) | 0.525 |
| **Family work balance** | | | | | | |
| Yes | Ref | | Ref | | Ref | |
| No | 1.29(0.63–2.65) | 0.480 | 0.76(0.37–1.54) | 0.447 | 0.63(0.29–1.41) | 0.262 |
| **Hostile work environment** | | | | | | |
| No | Ref | | Ref | | Ref | |

*(Continued)*

**Table 3.** (Continued)

| Variables | Depression symptoms | | Anxiety symptoms | | Stress symptoms | |
|---|---|---|---|---|---|---|
| | AOR (95%CI) | p value | AOR (95%CI) | p value | AOR (95%CI) | p value |
| Yes | 0.93(0.39–2.24) | 0.875 | 0.6(0.25–1.44) | 0.254 | 0.68(0.26–1.75) | 0.418 |

CI: Confidence interval; COR: Crude Odds Ratio; AOR: Adjusted Odds

Ref: Reference group

# Internal Finish Worker, Armature Fixing, Painter and Electrician

## Kami, Damai, Badi, Gaine, Madhesi Brahmin, Madhesi Janajati

[22] prevalence of depression among construction workers of South Korea. Similarly, another study by Boschman et al. among Dutch bricklayers showed 18% prevalence of depression which were higher than our findings [23]. Two studies from the United Kingdom [24] and South Korea [21] reported the prevalence of anxiety to be 48% and 43%, respectively, which were higher compared to our findings. This difference in prevalence compared to our study might be due to the difference in tools used to assess depression and anxiety symptoms and the difference in study location and population. In addition, there are differences in working structure, facilities and policies among high-income and low-income countries [25]. More than a quarter of suicides in the construction industry have been linked to an untreated or undiagnosed mental disease, according to previous studies [5, 26]. Majority of people with depression and/or anxiety disorder do not think about, let alone attempt or commit suicide [27]. The risk of suicide among construction workers can be decreased through early detection, treatment, and prevention of depression and anxiety [28].

In this study, we found that depression symptoms were positively associated with poor sleep quality indicating a higher likelihood of depression among construction workers with poor sleep quality. This finding is consistent with prior studies [29–31]. A study by Kim et al. in Korea reported that workers who had a poor quality of sleep were more likely to feel depressed and fatigued than people who had a good quality of sleep [30]. Similarly, another study conducted among male construction workers revealed that 44.3% of depression was linked to occupational stress, exhaustion, and sleep quality [29]. The results of a study conducted in Japan indicate a potential link between the severity of insomnia and increased risk of developing depression in the future [32].

The Brahmin ethnic group, which is considered a high-status group in the Nepalese community, demonstrated a higher prevalence of stress symptoms in comparison to other ethnic groups and revealed a positive association between being Brahmin and experiencing stress symptoms. This positive association between Brahmin ethnicity and stress symptoms contradicts previous research in Nepal [33, 34] which found higher rates of mental health issues among lower minority groups compared to high castes. One possible explanation for the lower rates of stress symptoms among other ethnic groups is under-reporting due to the stigma surrounding mental illness, particularly in low-income countries such as Nepal [35].

Current smokers showed a high prevalence of stress. Many studies have found that stress is a barrier to quitting smoking, and that persons who are less stressed have a better probability of quitting successfully [36, 37]. Psychosocial factors, such as perceived stress, are thought to modulate the association between socioeconomic status and smoking [38, 39].

Research by Kang et al. identified five genetic markers that are potentially associated with an increased risk of depressive disorder in females compared to males [40]. A meta-analysis among the 19639 patient population showed males were 0.63 times less likely to have

depression compared to females [41] but our study failed to show significant association between depression symptoms and sex of the construction workers.

Job insecurity triggers financial and relational strain and initiates worries of job loss which leads to depression, anxiety and stress and even suicide [42]. Our study, however did not find any association of job insecurity with depression, anxiety and stress symptoms. While a meta-analysis found a strong association between job satisfaction and decreased depression [43], we were unable to determine any such association in our study among construction workers.

This study has several strengths. First, it is among the first studies to identify the prevalence of depression symptoms, anxiety, and stress symptoms among construction workers in Nepal. Second, we used the Nepalese version DASS-21 for assessing depression symptoms, anxiety, and stress which is a validated tool and has accepted Cronbach's alpha values. Third, use of a well-validated electronic form for data collection, regular data checks for errors and timely cleaning of collected data.

However, this study has some limitations. First, this study was carried out in central Nepal using a convenience sampling technique so the result may not be generalized to overall Nepal. Second, the cross-sectional nature of the study prevents the directionality of the association between depression, anxiety and stress symptoms, and different predisposing factors. Lastly, we used one item for the measurement of each psychosocial domain such as job satisfaction, job insecurity, hostile work environment and family work balance which might result in low reliability and validity for each domain.

## Conclusions

The study found a high prevalence of depression, anxiety, and stress symptoms among construction workers. To address this, it is recommended to develop and implement evidence-based and appropriate community-based mental health prevention programs by involving community members and construction workers to take into account their unique needs and challenges.

## Acknowledgments

We would like to acknowledge all participants who provided time for our study. We would like to express our gratitude to Saugat Pratap KC and Bipul Lamichhane from HERD International for their support in correcting the grammar and language used in our manuscript. Additionally, we are thankful to all those who provided us with direct and indirect support during the course of our study.

## Author Contributions

**Conceptualization:** Bikram Adhikari, Rajendra Karkee, Anup Ghimire.

**Data curation:** Bikram Adhikari.

**Formal analysis:** Bikram Adhikari.

**Funding acquisition:** Bikram Adhikari.

**Investigation:** Bikram Adhikari, Anup Ghimire.

**Methodology:** Bikram Adhikari, Nabin Adhikari, Bhawana Shrestha, Rajendra Karkee, Anup Ghimire.

**Project administration:** Bikram Adhikari, Anup Ghimire.

**Resources:** Bikram Adhikari, Lisasha Poudel, Niroj Bhandari, Nabin Adhikari, Bhawana Shrestha, Bikram Poudel, Anupama Bishwokarma, Bihari Saran Kuikel, Dinesh Timalsena, Bandana Paneru, Minani Gurung, Pramesh Koju, Rajendra Karkee, Anup Ghimire.

**Software:** Bikram Adhikari, Bikram Poudel.

**Supervision:** Minani Gurung, Rajendra Karkee, Anup Ghimire.

**Validation:** Rajendra Karkee, Anup Ghimire.

**Visualization:** Bikram Adhikari, Anupama Bishwokarma, Bihari Saran Kuikel, Dinesh Timalsena, Bandana Paneru, Pramesh Koju.

**Writing – original draft:** Bikram Adhikari, Lisasha Poudel, Niroj Bhandari, Nabin Adhikari, Bhawana Shrestha, Bikram Poudel, Anupama Bishwokarma, Bihari Saran Kuikel, Dinesh Timalsena, Bandana Paneru, Minani Gurung, Pramesh Koju, Rajendra Karkee, Anup Ghimire.

**Writing – review & editing:** Bikram Adhikari, Lisasha Poudel, Niroj Bhandari, Nabin Adhikari, Bhawana Shrestha, Bikram Poudel, Anupama Bishwokarma, Bihari Saran Kuikel, Dinesh Timalsena, Bandana Paneru, Minani Gurung, Pramesh Koju, Rajendra Karkee, Anup Ghimire.

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
