## [Decision Letter · Decision Letter 0]

26 Sep 2022

PONE-D-22-12690Prevalence and factors associated with depression, anxiety and stress symptoms among construction workers of NepalPLOS ONE

Dear Dr. Adhikari,

Thank you for submitting your manuscript to PLOS ONE. After careful consideration, we feel that it has merit but does not fully meet PLOS ONE’s publication criteria as it currently stands. Therefore, we invite you to submit a revised version of the manuscript that addresses the points raised during the review process.

We look forward to receiving your revised manuscript.

Kind regards,

Shyam Sundar Budhathoki

Academic Editor

PLOS ONE

Journal Requirements:

Reviewers' comments:

Reviewer's Responses to Questions

**Comments to the Author**

1. Is the manuscript technically sound, and do the data support the conclusions?

Reviewer #1: Partly

Reviewer #2: Yes

2. Has the statistical analysis been performed appropriately and rigorously? 

Reviewer #1: No

Reviewer #2: Yes

3. Have the authors made all data underlying the findings in their manuscript fully available?

Reviewer #1: No

Reviewer #2: Yes

4. Is the manuscript presented in an intelligible fashion and written in standard English?

Reviewer #1: No

Reviewer #2: Yes

5. Review Comments to the Author

Reviewer #1: This is a cross-sectional study conducted with construction workers of Nepal. The authors aim to assess the prevalence of depressive, anxiety and stress symptoms and their association with socio-demographic, life-style and occupational factors among construction workers of Kavre district, Nepal. The biggest limitation of the study is the convenience sampling which makes it difficult to infer current conclusions. I have a number of suggestions below:

1. I recommend the objective be rephrased as “…..to assess psychological distress (depression, anxiety and stress) and associated factors among construction workers in Kavrepalanchok district, Nepal” instead of “…..to assess the prevalence of depressive, anxiety and stress symptoms and their association with socio-demographic, life-style and occupational factors among construction workers of Nepal. ”

2. Please provide references for lines 83-87.

3. Methods section should be more detailed. For example, how was the study area selected, how many wards were there in each municipality and how many were targeted for data collection, and who collected the data, etc.

4. Authors defined construction workers as members of specialist trades such as builders, electricians, carpenters, bricklayer, manual labor, armature fixing workers, internal finish workers and plumbers in the introduction. As stated in the inclusion criteria, is it only building construction workers who were interviewed in the present study? Please clarify.

5. Could the authors explain how current smokers and current drinker were defined in the study? How did the authors decide which risk factors to investigate in this study? I suggest writing a separate paragraph explaining the variables that were assessed.

6. Throughout the manuscript, I suggest the authors use either Kavre or its full name.

7. Please correct “convenient” to “convenience” on line 116.

8. Could the authors explain why they sought ethical approval from B.P. Koirala Institute of Health Sciences sought rather than the NHRC?

9. Please correct “life-style” to “lifestyle” on line 135 and throughout the manuscript.

10. Authors mentioned interquartile range on line 174 but it does not appear anywhere else in the manuscript. Is there a reason to mention it?

11. Results: What was the response rate?

12. The authors reported Tamang, Bhote, and Sherpa as ethnic groups on line 180 but this classification was not shown in the tables. It would be helpful if the authors could provide an explanation of how the classification of ethnicity was made and if applicable, the reference for it.

13. Could the authors explain how was the multivariate logistic regression performed and why?

14. Odds ratios in univariate and multivariate analyses are not correctly interpreted. I suggest the authors revise it.

15. Could the authors provide a reference to Kobotoolbox?

16. Line 273, 'Nepalease' should be changed to 'Nepalese,' and 'Vrahamin' should be changed to 'Brahmin,' as well. Line 282: please spell socioeconomic correctly.

17. Tables description should explain all the symbol in the table body like CI, n, COR, AOR etc.

18. Line 234, I suggest that authors also include the digits following the decimal.

19. Discussion: What could be the future research direction after this study?

20. Lines 303-305: please complete the sentences.

21. Authors should also expand the limitations and biases associated with the sampling strategy.

22. The recommendation ‘to develop and implement evidence-based and appropriate community-based mental health prevention programs for laborers and construction workers’ needs some further explanation.

23. Attaching a STROBE checklist would strengthen your study.

24. There were several grammatical errors in the manuscript. The authors should use professional editing services to edit the manuscript.

Reviewer #2: In the present work, the authors preformed the study on mental health of construction workers of Nepal and the topics is relevant to current context. This is a study of a neglected population and on sensitive topics, however, the manuscript needs revision before it is published in the journal. There is concerns multicollinearity issues and revision of tables of the result section. References need to be revised as par journal.

Abstract

Methods

Please explained why the study is community based cross sectional study. Is it necessary to add the process of ethical review in the abstract section?

Results

Please add adjusted OR when presenting the results section.

Introduction

Line 71-72, what does the information (International Labor Office in Nepal 2005) means?

Line 80, there are some error, please revise it.

Methods

I am not sure why the study is community based cross sectional study.

The author calculates the sample size of 284, however the author use 402 construction workers. It may create the confusion among the readers. So, it is suggested to revise this sentence.

In the statistical analysis section, it is suggested to do multicollinearity before doing the logistic regression whether to check whether there is correlation of the variables.

Results

It is suggested to use respondents or construction workers instead of participants.

In Table 1, it is suggested to write the number in the relevant section as well.

It is also suggested to merge castes into less category.

It is recommended to revise the table. Please make the logistic table showing the results of crude and adjusted odd ratio of depressive symptoms, similar table for anxiety and similar table for stress

It is suggested to revise the analysis if relevant. Please do the multivariate logistic regression analysis of the variable that was found to be significant in univariate logistic regression analysis.

Discussion

Please compare the prevalence of depression, anxiety and stress with National Mental Health Survey, Nepal 2020 that many reflect whether the prevalence is high, moderate and low.

Line 246-249, please revised the writing style

Line 265-266, please revise the writing style

Line 288-298, please review whether this sentence is importance and relevant to the manuscript.

References

Please revise the reference and follow the referencing style based on the journal requirement. In the reference section, some journals are in full name and some journals are in abbreviations.

6. PLOS authors have the option to publish the peer review history of their article (what does this mean?). If published, this will include your full peer review and any attached files.

Reviewer #1: No

Reviewer #2: No

---

## [Author Response · Author response to Decision Letter 0]

25 Jan 2023

Respected reviewers,

I would like to thank all reviewers for their wonderful feedback. The response to the reviewer’s comment are as follows:

Reviewer #1: This is a cross-sectional study conducted with construction workers of Nepal. The authors aim to assess the prevalence of depressive, anxiety and stress symptoms and their association with socio-demographic, life-style and occupational factors among construction workers of Kavre district, Nepal. The biggest limitation of the study is the convenience sampling which makes it difficult to infer current conclusions. I have a number of suggestions below:

1. I recommend the objective be rephrased as “…..to assess psychological distress (depression, anxiety and stress) and associated factors among construction workers in Kavrepalanchok district, Nepal” instead of “…..to assess the prevalence of depressive, anxiety and stress symptoms and their association with socio-demographic, life-style and occupational factors among construction workers of Nepal. ”

Answer: We have corrected as per your suggestion.

2. Please provide references for lines 83-87.

Answer: Reference is added to the line 83-87.

3. Methods section should be more detailed. For example, how was the study area selected, how many wards were there in each municipality and how many were targeted for data collection, and who collected the data, etc.

Answer: the method sections are changed as per your suggestion by providing details you expected.

4. Authors defined construction workers as members of specialist trades such as builders, electricians, carpenters, bricklayer, manual labor, armature fixing workers, internal finish workers and plumbers in the introduction. As stated in the inclusion criteria, is it only building construction workers who were interviewed in the present study? Please clarify.

Answer: It is clarified 

5. Could the authors explain how current smokers and current drinker were defined in the study? How did the authors decide which risk factors to investigate in this study? I suggest writing a separate paragraph explaining the variables that were assessed.

Answer: It is explained in the current version of manuscript.

6. Throughout the manuscript, I suggest the authors use either Kavre or its full name.

Answer: Done

7. Please correct “convenient” to “convenience” on line 116.

Answer: Done

8. Could the authors explain why they sought ethical approval from B.P. Koirala Institute of Health Sciences sought rather than the NHRC?

Answer: This manuscript is the part of thesis carried out in BPKIHS. The IRC of BPKIHS is the body of NHRC to which NHRC have given certain rights to provide ethical approval for observational studies including cross-sectional studies.

9. Please correct “life-style” to “lifestyle” on line 135 and throughout the manuscript.

Answer: done

10. Authors mentioned interquartile range on line 174 but it does not appear anywhere else in the manuscript. Is there a reason to mention it?

Answer: The comment is addressed.

11. Results: What was the response rate?

Answer: The response rate is added to first part of result section.

12. The authors reported Tamang, Bhote, and Sherpa as ethnic groups on line 180 but this classification was not shown in the tables. It would be helpful if the authors could provide an explanation of how the classification of ethnicity was made and if applicable, the reference for it.

Answer: The classification of the ethnicity is explained in the method section. 

13. Could the authors explain how was the multivariate logistic regression performed and why?

All variables from univariate analysis were kept in the multivariate logistic regression model. 

Answer: All variables in univariate logistic regression were included in multivariate logistic regression analysis.

14. Odds ratios in univariate and multivariate analyses are not correctly interpreted. I suggest the authors revise it.

Answer: We revised the interpretation of univariate and multivariate 

15. Could the authors provide a reference to Kobotoolbox?

Answer: reference is added to kobotoolbox

16. Line 273, 'Nepalease' should be changed to 'Nepalese,' and 'Vrahamin' should be changed to 'Brahmin,' as well. Line 282: please spell socioeconomic correctly.

Answer: Changed

17. Tables description should explain all the symbol in the table body like CI, n, COR, AOR etc.

Answer: Added 

18. Line 234, I suggest that authors also include the digits following the decimal.

Answer: Digits following decimal are added in line 234

19. Discussion: What could be the future research direction after this study?

Answer: It is added now in the manuscript

20. Lines 303-305: please complete the sentences.

Answer: Edited

21. Authors should also expand the limitations and biases associated with the sampling strategy.

Answer: A limitation related to the sampling strategy is added to the manuscript.

22. The recommendation ‘to develop and implement evidence-based and appropriate community-based mental health prevention programs for laborers and construction workers’ needs some further explanation.

Answer: This comment is addressed.

23. Attaching a STROBE checklist would strengthen your study.

Answer: Strobe guideline is followed but not attached here with this manuscript 

24. There were several grammatical errors in the manuscript. The authors should use professional editing services to edit the manuscript.

Answer: We corrected grammatical mistakes that we found in the manuscript.

Reviewer #2: In the present work, the authors preformed the study on mental health of construction workers of Nepal and the topics is relevant to current context. This is a study of a neglected population and on sensitive topics, however, the manuscript needs revision before it is published in the journal. There is concerns multicollinearity issues and revision of tables of the result section. References need to be revised as par journal.

Answer: Multicollinearity testing was conducted to check for the multicollinearity issue. The variance inlation factor (VIF) was used to identify multicollinear. All variables have VIF less than 2.5. The tables are kept to match the need of journal and to prevent longer tables, regression tables are splitted to univariate and multivariate tables.

Abstract

Methods

Please explained why the study is community based cross sectional study. Is it necessary to add the process of ethical review in the abstract section?

Ans: The comment is addressed

Results

Please add adjusted OR when presenting the results section.

Ans: corrected

Introduction

Line 71-72, what does the information (International Labor Office in Nepal 2005) means?

Ans: this issue is resolved 

Line 80, there are some error, please revise it.

Ans: the issue is resolved 

Methods

I am not sure why the study is community based cross sectional study.

Ans: The study was conducted directly in the community so it’s a community based crossectional study but as per your comment and prevent confusion, only crossectional study is kept as study design

The author calculates the sample size of 284, however the author use 402 construction workers. It may create the confusion among the readers. So, it is suggested to revise this sentence.

Ans: The statement is revised 

In the statistical analysis section, it is suggested to do multicollinearity before doing the logistic regression whether to check whether there is correlation of the variables.

Ans: Multicolinearlity testing was conducted before selecting variables in the model 

Results

It is suggested to use respondents or construction workers instead of participants.

In Table 1, it is suggested to write the number in the relevant section as well.

It is also suggested to merge castes into less category.

Ans: The caste was reduced to 4 categories from larger number of categories and cant be merged because with merging and reducing categories will result in loss of information.

It is recommended to revise the table. Please make the logistic table showing the results of crude and adjusted odd ratio of depressive symptoms, similar table for anxiety and similar table for stress

With this suggestion the count of tables will increase which is not recommended as per journal requirement so we planned to keep two tables, one with univariate analysis and another with multivariate analysis.

It is suggested to revise the analysis if relevant. Please do the multivariate logistic regression analysis of the variable that was found to be significant in univariate logistic regression analysis.

Ans: Since it is not aimed to generate prediction model, we kept relevant variables to explore association between outcome and independent variables 

Discussion

Please compare the prevalence of depression, anxiety and stress with National Mental Health Survey, Nepal 2020 that many reflect whether the prevalence is high, moderate and low.

Ans: The National Mental Health Survey, Nepal 2020 have not determined depression, anxiety and stress symptoms so we could not compare our findings with National Mental Health Survey, Nepal 2020.

Line 246-249, please revise the writing style

Answer: The line 246-249 is revised for clearness

Line 265-266, please revise the writing style

Ans: Line 265-266 is rephrased 

Line 288-298, please review whether this sentence is importance and relevant to the manuscript.

Ans: Line 288-298 is corrected

References

Please revise the reference and follow the referencing style based on the journal requirement. In the reference section, some journals are in full name and some journals are in abbreviations.

Ans: All references are corrected as per the referencing style of the journal 

Thank you very much for your valuable feedback. It was very helpful to improve this manuscript. 

Best Regards

Bikram Adhikari

---

## [Decision Letter · Decision Letter 1]

20 Feb 2023

PONE-D-22-12690R1Prevalence and factors associated with depression, anxiety and stress symptoms among construction workers in NepalPLOS ONE

Dear Mr. Adhikari,

Thank you for submitting your manuscript to PLOS ONE. After careful consideration, we feel that it has merit but does not fully meet PLOS ONE’s publication criteria as it currently stands. Therefore, we invite you to submit a revised version of the manuscript that addresses the points raised during the review process.

We look forward to receiving your revised manuscript.

Kind regards,

Shyam Sundar Budhathoki

Academic Editor

PLOS ONE

Journal Requirements:

Reviewers' comments:

Reviewer's Responses to Questions

**Comments to the Author**

1. If the authors have adequately addressed your comments raised in a previous round of review and you feel that this manuscript is now acceptable for publication, you may indicate that here to bypass the “Comments to the Author” section, enter your conflict of interest statement in the “Confidential to Editor” section, and submit your "Accept" recommendation.

Reviewer #1: (No Response)

Reviewer #3: (No Response)

2. Is the manuscript technically sound, and do the data support the conclusions?

Reviewer #1: Yes

Reviewer #3: Yes

3. Has the statistical analysis been performed appropriately and rigorously? 

Reviewer #1: Yes

Reviewer #3: Yes

4. Have the authors made all data underlying the findings in their manuscript fully available?

Reviewer #1: No

Reviewer #3: (No Response)

5. Is the manuscript presented in an intelligible fashion and written in standard English?

Reviewer #1: Yes

Reviewer #3: Yes

6. Review Comments to the Author

Reviewer #1: I appreciate your thorough responses to reviewer questions and concerns. In my previous comment, I suggested that authors rephrase the objective as “…..to assess psychological distress (depression, anxiety and stress) and associated factors among construction workers in Kavre district, Nepal”. The authors stated in their response that they have changed as per the suggestion; however, Kavre has not been mentioned in the objectives of the abstract and introduction. Results, 41.3% is not a majority, revise the wording. Besides these points, the authors have addressed my comments.

Reviewer #3: This is a cross-sectional study conducted among 402 construction workers in Banepa and Panauti municipalities of Kavre district, Nepal aimed to assess psychological distress (depression, anxiety and stress symptoms) and its association with socio-demographic, lifestyle, and occupational factors among construction workers in Nepal. I have mentioned few suggestions below:

Abstract

1. Introduction: In line 35, i suggest authors to use aimed instead of aims.

2. Introduction: From line 86-94, same reference 6 has been used by the author. I recommend author to add any other reference article with similar information if its available. I suggest the authors to write the objectives as written in the abstract section: to assess the prevalence of depression, anxiety and stress symptoms and its association with socio-demographic, lifestyle and occupational factors among construction workers in Nepal.

3. Methods: I recommend authors to mention the study as cross-sectional study. community based is not required to be mentioned.

4. Data collection: Could the authors explain regarding translation of english questionnaire into the local language and back-translation into English. I would suggest to provide information regarding pre-testing if it was performed. In line 148, 'b' is to be added in 'uddhism'.

5. Line 177, mention as 'depression, anxiety and stress symptoms'.

6. In line 196, for statistical analysis authors mentioned use of 'SPSS version 20' whereas in abstract section in line 45, its mentioned 'R version 3.6.2'. Please clarify.

7. Line 217, 61.25% is missing.

8. In Univariate and Multivariate analysis table, what does 1 denote? You may also signify it as Ref. Mention at the end of table, Ref: Reference group if relevant.

9. Line 221, its better to use 'depression' than 'depressive'.

10. In description of Table 2, authors should also add regarding ethnicity and stress symptoms as it is significant as shown in table.

11. Tables description should explain only the symbol used in the table body like CI, COR or AOR etc. Authors can also mention Title of the table and in bracket total study population as (N=402).

12. There are grammatical errors in the manuscript. The authors are requested to go through it and edit the manuscript where required.

13. I suggest authors to revise the references as per the referencing style based on the journal. Also fonts of the text should be similar. Please consider.

7. PLOS authors have the option to publish the peer review history of their article (what does this mean?). If published, this will include your full peer review and any attached files.

Reviewer #1: No

Reviewer #3: No

---

## [Author Response · Author response to Decision Letter 1]

6 Mar 2023

Thank you to all reviewers and editors for providing constructive feedback and comments. I’ve addressed all comments that you have provided in the manuscript and submitted it again after careful revision of grammar and errors.

Reviewer #1: I appreciate your thorough responses to reviewer questions and concerns. In my previous comment, I suggested that authors rephrase the objective as “…..to assess psychological distress (depression, anxiety and stress) and associated factors among construction workers in Kavre district, Nepal”. The authors stated in their response that they have changed as per the suggestion; however, Kavre has not been mentioned in the objectives of the abstract and introduction. Results, 41.3% is not a majority, revise the wording. Besides these points, the authors have addressed my comments.

Ans: Thank you reviewer 1, I’ve addressed this comment in the new submitted manuscript.

Reviewer #3: This is a cross-sectional study conducted among 402 construction workers in Banepa and Panauti municipalities of Kavre district, Nepal aimed to assess psychological distress (depression, anxiety and stress symptoms) and its association with socio-demographic, lifestyle, and occupational factors among construction workers in Nepal. I have mentioned few suggestions below:

Abstract

1. Introduction: In line 35, i suggest authors to use aimed instead of aims.

Ans: Corrected

2. Introduction: From line 86-94, same reference 6 has been used by the author. I recommend author to add any other reference article with similar information if its available. I suggest the authors to write the objectives as written in the abstract section: to assess the prevalence of depression, anxiety and stress symptoms and its association with socio-demographic, lifestyle and occupational factors among construction workers in Nepal.

Ans: Corrected

3. Methods: I recommend authors to mention the study as cross-sectional study. community based is not required to be mentioned.

Ans: Corrected

4. Data collection: Could the authors explain regarding translation of english questionnaire into the local language and back-translation into English. I would suggest to provide information regarding pre-testing if it was performed. In line 148, 'b' is to be added in 'uddhism'.

Ans: Corrected

5. Line 177, mention as 'depression, anxiety, and stress symptoms'.

Ans: Corrected

6. In line 196, for statistical analysis authors mentioned use of 'SPSS version 20' whereas in abstract section in line 45, its mentioned 'R version 3.6.2'. Please clarify.

It was a typo, We used R version 3.6.2 for statistical analysis. Thank you addressing this serious issue

7. Line 217, 61.25% is missing.

8. In Univariate and Multivariate analysis table, what does 1 denote? You may also signify it as Ref. Mention at the end of table, Ref: Reference group if relevant.

Ans: Changed and annotated

9. Line 221, its better to use 'depression' than 'depressive'.

Ans: Corrected

10. In description of Table 2, authors should also add regarding ethnicity and stress symptoms as it is significant as shown in table.

Ans: Corrected

11. Tables description should explain only the symbol used in the table body like CI, COR or AOR etc. Authors can also mention Title of the table and in bracket total study population as (N=402).

Ans: Corrected

12. There are grammatical errors in the manuscript. The authors are requested to go through it and edit the manuscript where required.

Ans: Corrected

13. I suggest authors to revise the references as per the referencing style based on the journal. Also fonts of the text should be similar. Please consider.

Ans: Corrected

---

## [Editor Report · Decision Letter 2]

6 Apr 2023

Prevalence and factors associated with depression, anxiety and stress symptoms among construction workers in Nepal

PONE-D-22-12690R2

Dear Dr. Adhikari,

We’re pleased to inform you that your manuscript has been judged scientifically suitable for publication and will be formally accepted for publication once it meets all outstanding technical requirements.

Kind regards,

Shyam Sundar Budhathoki

Academic Editor

PLOS ONE

Additional Editor Comments (optional):

The manuscript will improve its readability by proof reading and ediing for english language. Please find a fluent academic english user if possible.
---

## [Editor Report · Acceptance letter]

19 May 2023

PONE-D-22-12690R2 

Prevalence and factors associated with depression, anxiety and stress symptoms among construction workers in Nepal 

Dear Dr. Adhikari:

I'm pleased to inform you that your manuscript has been deemed suitable for publication in PLOS ONE. Congratulations! Your manuscript is now with our production department. 

Kind regards, 

on behalf of

Dr. Shyam Sundar Budhathoki 

Academic Editor

PLOS ONE